# SEQ-rPPG: A Fast BVP Signal Extraction Method From Frame Sequences

## Abstract

Remote photoplethysmography (rPPG) can be widely used in various kinds of physical, health, and emotional monitoring, such as monitoring the heart rate of drivers, consumers, the elderly, and infants. Several rPPG methods have been proposed in the past few years, but non-contact heart rate estimation in realistic situations is still challenging. It is observed that the existing deep learning-based rPPG methods can not achieve real-time performance on low-cost devices. To deal with this problem, a simple, fast, and pre-processing-free approach called sequence-based rPPG (SEQ-rPPG) is proposed for non-contact heart rate estimation. SEQ-rPPG first transforms the RGB frame sequence into the new signal sequence by learning-based linear mapping and then outputs the final BVP signal using 1DCNN-based spectral transform and time-domain filtering. It requires no complex pre-processing, has the fastest speed, can run in real-time on mobile ARM CPUs, and can achieve real-time beat-to-beat performance on desktop CPUs. Furthermore, We present a well-annotated dataset, focusing on constructing a large-size and highly synchronized PPG and video. The entire data set will be made available to the research community. Benefiting from this high-quality dataset, other deep learning-based models reduced errors. To prove the efficacy of the proposed method, the comparison is done with state-of-the-art methods. The experimental results on both self-build and publicly available datasets have demonstrated the effectiveness of the proposed method. We also verified that the processing in the frequency domain is effective.

## 1 Introduction

The Blood volume pulse (BVP) is a physiological measurement used to extract physiological signals from the heart. The most used method of BVP signal extraction is photoplethysmogram (PPG). However, PPG requires the subject to wear an optical contact sensor, which can cause discomfort and lead to difficulties, such as monitoring patients in pediatric intensive care units. Non-contact BVP extraction is possible via high-sensitivity cameras and webcams using ambient light as a source of illumination (Takano & Ohta, 2007; Verkruysse et al., 2008). So remote PPG (rPPG) has attracted extensive attention in recent years.

Early rPPG methods mainly focused on temporal modeling. For example, independent component analysis (ICA) Poh et al. (2010a) is used for extracting BVP signals from RGB signals. Furthermore, Poh et al. (2010b) added the detrending operation to improve the robustness of ICA. But the blind source separation techniques in RGB color space show limited success. de Haan & Jeanne (2013) propose a chrominance-based method that is robust to luminance. We can find that temporal modeling focused on linear optical models and filter-based post-processing, which empirically proved to be fast and effective. In addition, spatial modeling is also an important way to extract BVP Tulyakov et al. (2016); Bobbia et al. (2019). They attempt to select the part of the face with the highest signal-to-noise ratio from different regions of the face. This way, it can accommodate more head movements, illumination, and shading. However, these handcrafted algorithms with poor accuracy compared to deep learning-based approaches.

Recently, deep learning-based approaches (Chen & McDuff, 2018; Niu et al., 2020a;b; Liu et al., 2020; Song et al., 2021; Yu et al., 2019; Lu et al., 2021; Liu et al., 2021; Yu et al., 2022) have achieved better accuracy. However, the computation cost of these methods is much larger than

handcrafted algorithms, making it difficult to deploy on low-cost devices. The high computation cost comes from two sources. First is the pre-processing. In Niu et al. (2020a;b); Song et al. (2021); Lu et al. (2021), algorithms require fine segmentation of the face, so these algorithms rely on handcrafted features, and the computation cost required for pre-processing is even greater than the model itself. Second, some end-to-end models(Liu et al., 2021; Yu et al., 2022) achieve high accuracy, but the models are too large to run on devices without GPUs. Although some models (Yu et al., 2019; Liu et al., 2020) can run in real-time on mobile CPUs with good accuracy, the computation cost is still too high. rPPG is commonly used in affective computing and telemedicine, where users must juggle other components while running rPPG applications. It is unacceptable to allocate most of the computation resources to real-time rPPG. Therefore, designing an rPPG algorithm that only takes up a few computation resources is necessary.

Apart from the algorithm design, the dataset is also vital for obtaining better learning models. Different training sets have a huge impact on model performance, the main reason is that most of the datasets are not well synchronized with the video signal and the BVP signal. There are some datasets that are highly synchronized, such as PURE, SCAMPS(Stricker et al., 2014; McDuff et al., 2022). However, PURE only has 59 minutes of video, and SCAMPS has not yet shown strong enough generalization as a synthetic dataset. In addition, some works (Yu et al., 2019; Botina-Monsalve et al., 2022; Comas et al., 2022) focused on making the model adaptable to non-synchronized datasets, a more common practice is to use unsupervised methods to synchronize signals (eg. POS), but it is still an open problem to design methods robust to non-synchronization. Although there are different ways to alleviate the problem of non-synchronized, the scarcity of high-quality datasets still limits the performance of some models.

To deal with the problems mentioned above, an effective BVP extraction method is proposed, and a new dataset is constructed. The main contributions of this paper can be summarized as follows:

- A simple, fast, and pre-processing-free method is proposed for non-contact BVP extraction. The proposed method, called sequence-based rPPG (SEQ-rPPG), uses linear mapping, spectral transform, and time-domain filtering to extract BVP signals. The proposed method requires no complex pre-processing and can run on mobile ARM CPUs in real-time. Its computation cost and memory usage are much smaller than existing algorithms.

- A large, synchronized dataset is built for better model training. In this paper, we developed software that simultaneously obtains the ground truth signal from the blood pulse meter and the RGB signal from the webcam, and all UNIX timestamps are recorded. It is designed for large-scale training, more than 30 hours (3.24M frames) of video available, and larger in duration than any public dataset.

## 2 RELATED WORK

### 2.1 END-TO-END PHYSIOLOGICAL SIGNAL MEASUREMENT NETWORK

Many studies focus on simplifying the pre-processing process or even removing it and are vague in their end-to-end definitions, and they all claim to be end-to-end models. In DeepPhys (Chen & McDuff, 2018) and MTTS-CAN (Liu et al., 2020), the pre-processing algorithms generate differential and average frames, respectively, then enter them into the motion branch and appearance branch of the network. In EfficientPhys (Liu et al., 2021), they use a pre-processing approach similar to MTTS-CAN, yet within the model and determine the normalization parameters by learning. It is hard to say if this is a pre-processing or part of the model internals. In PhysNet (Yu et al., 2019) and PhysFormer (Yu et al., 2022), the model accepts original image input directly without additional pre-processing. MTTS-CAN is an improved version of DeepPhys, which has higher accuracy, and although they both require pre-processing, the pre-processing is simple and requires little additional cost. PhysFormer and PhysNet are both end-to-end models. However, the number of parameters and computation of PhysFormer is much larger than the latter, which is not enough to reflect the advantages of lightweight and real-time. So EfficientPhys, MTTS-CAN, and PhysNet are representative end-to-end models, and we will compare the proposed method with them.

## 2.2 FAST FOURIER CONVOLUTION

Chi et al. (2020) proposed Fast Fourier Convolution (FFC), which can obtain an extensive range of receptive fields at a shallow layer of the network. It improves accuracy on multiple tasks and datasets by replacing the convolutional layers in general convolutional networks with FFC layers. Suvorov et al. (2022) applies FFC to a wide range of image restoration, showing superior performance, especially for periodic images. Shchekotov et al. (2022) applied FFC to speech and audio signal denoising, significantly improving the baseline method and reducing the number of parameters. FFC is particularly sensitive to periodic signals and will significantly affect work that deals with periodic signals, such as audio noise reduction and rPPG. The difference between the above work and ours is that we do not model spatially and do not divide the time and frequency domain channels. We use simple time domain channels and skip them by residual connection. In this view, SEQ-rPPG does not model the video. We filter the noise from the 1D sequence and extract the BVP signal.

## 3 METHOD

Handcrafted algorithms based on separate reflection components (Shafer, 1985) assume that the BVP signal in the RGB image comes through a linear combination of different frequency rays, while the skin-reflected light contains a specular reflection component and a diffuse reflection component. In post-processing, stationary signals and noise are filtered, while periodic signals generated by fluctuations in hemoglobin concentration are passed. Color pixel's dichromatic reflection model is shown below:

$$C_{aX+bY} = a\boldsymbol{C}_X + b\boldsymbol{C}_Y, \tag{1}$$

where X and Y are power spectral density (PSD), PSD is the most commonly used method for heart rate extraction, which means that the rPPG signal can always be obtained by linear mapping of color space. In the early rPPG algorithm(Poh et al., 2010a;b; de Haan & Jeanne, 2013; Wang et al., 2017; Tulyakov et al., 2016), there are two steps from the RGB signal to the BVP signal. Firstly, based on a linear dimensionality reduction method to obtain a 1-dimensional signal, and secondly, apply filtering post-processing in the time domain or frequency domain to filter out noise. In this paper, SEQ-rPPG also performs two main steps (see Fig. 1): first, it linearly maps the raw 8x8 facial RGB frames to the original signal sequence. Then it uses 1D convolution to filter the original signal sequence and outputs the BVP waveform. The linear mapping is done inside the model, so this method does not require pre-processing. The parameters of the linear mapping module are obtained by learning, and after training, it will have the appropriate weights to combine the RGB signals.

### 3.1 LINEAR MAPPING MODULE

Given an RGB facial video input $\mathbf{X} \in \mathbb{R}^{N \times W \times H \times C}$, first transpose it to $\mathbf{X} \in \mathbb{R}^{N \times C \times W \times H}$, ensure the channel is in the second dimension, then reshape it to $\boldsymbol{X} \in \mathbb{R}^{NC \times WH}$. $\boldsymbol{X}$ consists of $N$ sets of RGB signals, the original signal is converted from RGB as

$$Y_n = \sum_{i=1}^{WH} \boldsymbol{m}_i \times (R_i, G_i, B_i)^T + b_i, \tag{2}$$

where $\boldsymbol{m}$ is a row vector, which maps RGB colors to real numbers, $b$ is the bias. The transformation of colors can be achieved by convolution without activation layers, it uses a linear convolution layer with kernel size = 3 and stride = 3, linear mapping outputs multi-channel sequences, the number of channels depends on the number of convolution kernels.

### 3.2 SIGNAL PROCESSING MODULE

For the given original signal, it is usually filtered using several convolutional layers. This filtering is performed in the time domain, which means that adjacent signal points will be strongly correlated while distant signal points will be weakly correlated. For periodic signals, considering the correlation between similar periods helps filter noise from a wider range. From the signal processing point of view, classical BVP signal extraction usually requires filters to deal with the noise. Time-domain filters include Gaussian filters, moving average filters, and detrending, etc. Frequency domain filters are commonly bandpass filters to extract signals at specific frequencies. In summary, classical

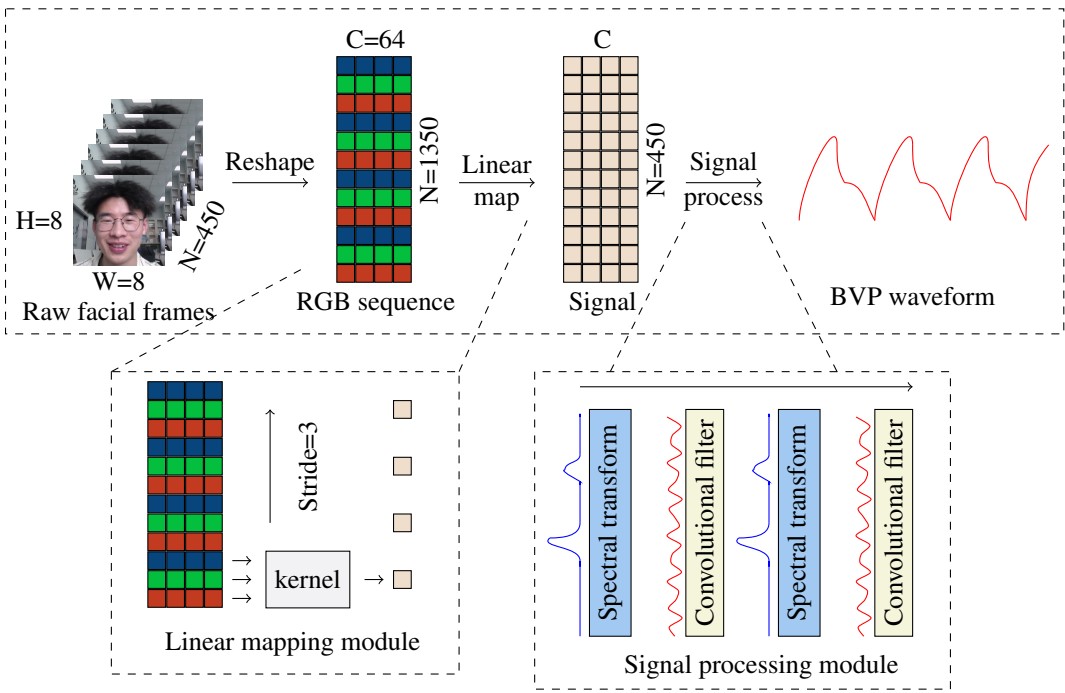

Figure 1: Overview of our framework.

methods contain alternating use of frequency domain filters and time domain filters. Inspired by the classical methods, we use the spectral transform layer to complete the processing in the frequency domain, while the traditional 1D convolution is used for the time domain processing (see Fig. 2). For a signal $\boldsymbol{Y} \in \mathbb{R}^{N \times C}$, the spectral filtering layer first performs a Real Fast Fourier Transform (RFFT) on each channel, obtain frequency domain signal $\boldsymbol{Y}_{Freq} \in \mathbb{Z}^{\lfloor \frac{N+1}{2} \rfloor \times C}$, then it is decomposed into a real part $\boldsymbol{Y}_{Real}$ and an imaginary part $\boldsymbol{Y}_{Img}$, they are combined on the channel as $\boldsymbol{Y}_{Comb} \in \mathbb{R}^{\lfloor \frac{N+1}{2} \rfloor \times 2C}$. Then a convolution layer is applied to it and re-decomposes the output into real and imaginary parts. It is converted to complex numbers and recovered to the time domain signal by Inverse Real Fast Fourier Transform (IRFFT). The output signal is mixed with the original signal through the residual connection, and the number of channels remains constant throughout the process. The final signal is obtained by alternating spectral transform and 1D convolution operation.

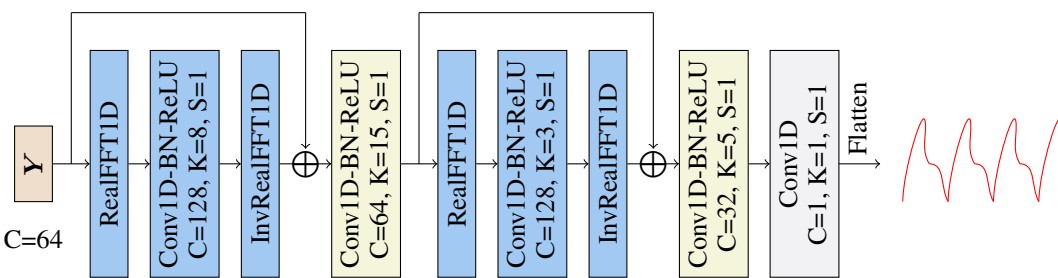

Figure 2: Signal processing module.

## 4 DATASET

To collect the dataset, we developed data capture software to obtain raw data from Logitech C930c webcam and Contech CMS50E Oximeter. The webcam has two capture modes, 1920*1080 resolution, MJPG encoding, and 640*480 resolution, YUY2 encoding. Each mode has a frame rate of

30fps. MJPG is a common fast intra-frame compression format, and YUY2 is a format for raw YUV images that uses 4 bytes to store 2 Y-components, 1 U-component, and 1 V-component. There are three video encoding formats, RAW RGB, MJPG, and H264. RAW RGB is the original lossless compressed RGB format, and H264 is the common inter-frame compression format on the Internet. Multiple formats are recorded simultaneously. Ground truth BVP waveform data is read from an oximeter, the sampling rate is 20 per second. During data recording, subjects were required to complete a series of tasks or watch a video, and subjects were not required to keep their heads stable, so that they may have experienced larger head movements. After completing the assigned task, the subject will take a short break, and the experimenter will assign him/her the next task. All 58 subjects (16 male and 42 female) are Chinese students, mostly master's students, and some girls may wear makeup. Excluding unavailable data due to formatting errors, incomplete tasks, or equipment failure, the video was available for over 30 hours (3.24M frames) of the video. Fig. 3 shows that the proposed dataset is well synchronized, where GT is the ground truth signal, and Video represents the BVP signal extracted from the video. More details of the self-build dataset can be found in Table 1 and Fig. 4. As shown in Fig. 4, this dataset contains head movement, illumination variation, and expression change.

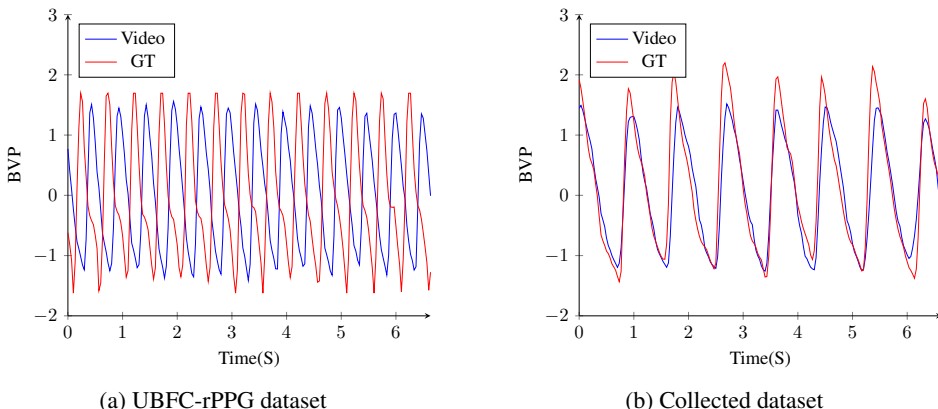

(a) UBFC-rPPG dataset          (b) Collected dataset

Figure 3: BVP signal extracted from UBFC-rPPG video and our collected video, where the collected dataset is highly synchronized.

Table 1: Data collection workflows

| Sub-dataset | Task or induction video | Duration(S) | Camera codec | Video codec | Resolution |
|---|---|---|---|---|---|
| rPPG | Relaxed | 120 | YUY2 | RGB,MJPG,H264 | 640×480 |
| | Relaxed & Dark | 120 | | | |
| | Play a game | 120 | | | |
| | Read an article | 120 | | | |
| Emotion | Natural scenery | 120 | MJPG | MJPG,H264 | 1920×1080 |
| | Puzzle game | 180 | | | |
| | Comedy | 120 | | | |
| | Illusion picture | 20 | | | |
| | Academic paper | 60 | | | |
| | Video about yawn | 60 | | | |
| Engagement | Video-based learning | 240 | MJPG | MJPG,H264 | 1920×1080 |
| | Textbook-based learning | 480 | | | |
| | Watch a public class | 420 | | | |

# 5 EXPERIMENTS

## 5.1 TRAINING DATA & TESTING DATA

Since our model cannot be trained effectively on non-synchronized datasets, we only trained it on the collected dataset and PURE dataset(Stricker et al., 2014), excluding a small amount of data with

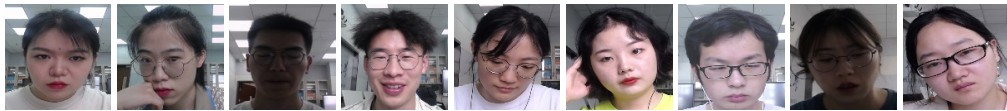

Figure 4: Overview of our dataset, subjects' faces can move unconstrained and emotions are induced during tasks.

a low frame rate or wrong format due to insufficient performance during recording. Although we acquired the data in multiple video encoding formats, only the MJPG format was used for the entire training and testing. For the self-build dataset, we randomly divided the 58 subjects into five folds, the first four folds were used as the training set, and the last fold was used as the testing set, and all videos of the same subject would only appear in the same fold. The test is divided into two parts, first on the whole testing set and then on the rPPG subset, which contains more complex scenarios with motion. Details about the dataset partitioning are saved in a CSV file and made public along with the dataset. Models were also tested on the UBFC-rPPG(Bobbia et al., 2019) and PURE public dataset. UBFC-rPPG includes 42 videos from 42 subjects of approximately 1 minute each, all recorded at a resolution of 640x480, a frame rate of 30 fps, in RAW RGB format, and all faces were kept stable. PURE includes 59 one-minute videos from 10 subjects, including 6 types of head movements, all recorded at a resolution of 640x480, a frame rate of 30 fps, in RAW RGB format.

## 5.2 IMPLEMENTATION & EXPERIMENT DETAILS.

To compare with Liu et al. (2020) and Yu et al. (2019) on the collected data set, we reproduced the TS-CAN directly using the source code provided by Liu et al. (2020). Since Tensorflow performance is usually higher than Pytorch, to be fair in performance tests, we implemented PhysNet and EfficientPhys using Tensorflow according to the details provided in Yu et al. (2019); Liu et al. (2021).

We have three models: SEQ-FT with alternating convolution in the frequency and time domains, SEQ-T with convolution in the time domain only, and the streamlined SEQ-tiny (see Fig 5). All models are trained and tested in the same Tensorflow environment, the version is 2.6.0, compiled with cuda 11.6, cudnn 8.3, AVX2. Hardware platforms include Nvidia Geforce RTX 3080 10G (GPU), AMD Ryzen9 5950x 16-core (CPU), ARM Cortex A72 4-core (CPU), when tested on ARM platform, also used Tensorflow 2.6.0, but does not include any acceleration support for GPUs or x86 CPUs. To train efficiently, we cut the video using a moving window with a step size of 3 seconds and used mediapipe to perform face detection in the middle of the frame (sometimes other people appeared in the background but were already processed). Finally, only the facial images were fed to the model.



Figure 5: Overview of SEQ-T & SEQ-tiny.

Although Liu et al. (2020); Yu et al. (2019); Liu et al. (2021) claims to be a pre-processing-free model, face detection had to be performed because of the large movements in the collected dataset. All models used the same data during training and testing, differing only in resolution. In addition, due to the lack of enough high heart rate samples in the collected dataset, we randomly selected a part of the segments with interval skipping half of the frames and did the same for the BVP waveform, which finally made the heart rate twice as high. The additional data accounted for about 25% and were evenly distributed in the training set. TS-CAN uses the Nadam optimizer with default parameters, MAE loss, batch size=128. PhysNet uses the SGD optimizer with learning rate=0.005, NP loss, batch size=32. our model uses the adam optimizer with default parameters, MSE loss, batch size=32. Classical methods and TS-CAN used a 0.75 to 2.5 Hz bandpass filters in the test. Neither our models nor PhysNet applied filters. CHROM,

POS and ICA(de Haan & Jeanne, 2013; Wang et al., 2017; Poh et al., 2010b) use the code provided in Boccignone et al. (2022) and Boccignone et al. (2020). The final heart rate is extracted from the peak of the signal power spectrum, ranging from 0.5 to 4 Hz. In calculating the error and Pearson correlation coefficients, we used a 30-second moving average window with 1-second step.

We evaluated the accuracy and speed of the model separately, where the accuracy metrics include Mean Absolute Error (MAE), Root Mean Square Error (RMSE), and Pearson correlation coefficient ($\rho$). For the speed test, we calculated the number of parameters, FLOPs (see Table 2), and memory usage of the model and tested the inference speed (ms) on desktop GPU, desktop CPU, and mobile CPU. We recorded the hardware load to indicate the efficiency of the model when the model was running at the highest speed.

Table 2: Theoretical parameters and FLOPs

| Model | Parameters | FLOPs |
|---|---|---|
| SEQ-tiny | 3.46K | 1.49M |
| SEQ-T | 152K | 136M |
| SEQ-FT | 266K | 157M |
| TS-CAN | 533K | 6.68G |
| PhysNet | 770K | 6.95G |
| EfficientPhys-C | 2.16M | 29.5G |

## 5.3 EXPERIMENTAL RESULTS

The extensive experimental results are presented in Tables 3, 4 and 5. The rPPG subset is more difficult because it has more light changes and facial activity. SEQ-FT achieved the best accuracy on the collected dataset, with second place going to EfficientPhys-C. The results show that SEQ-FT is more accurate than SEQ-T, and the spectral transformation is effective.

We compared the results of different models tested on UBFC-rPPG, which were trained on different training sets. Our models perform similarly, and all have high accuracy. Although the difference in the number of SEQ-FT and SEQ-tiny parameters is very large, their accuracy is similar, which may be caused by the stable head and stable lighting environment of UBFC-rPPG. TS-CAN and PhysNet are trained better on the collected dataset than on other datasets. In particular, the results on the collected dataset outperform those trained on UBFC-rPPG, indicating that a high-quality training set helps improve the model accuracy. Among all the cross-dataset results on UBFC-rPPG, the best is TS-CAN trained on the collected dataset, and the second best is PhysNet trained on the collected dataset. On the PURE dataset(Table 5), EfficientPhys-C achieves the best, followed by TS-CAN. Notably, compared to training on UBFC-rPPG, the improvement of EfficientPhys-C by training on the collected dataset is significant, with MAE dropping from 5.90 to 1.82.

As observed from the results, our model has very high accuracy on the collected dataset, so it is more suitable for educational scenarios. Our model is competitive with small models like TS-CAN on two public datasets, UBFC-rPPG and PURE, and our model parameters and FLOPs are much smaller. There is still a gap compared to larger models like EfficientPhys. The results of swapping the training set and the test set are interesting. Cross-dataset evaluations are usually trained on PURE and tested on UBFC-rPPG, but few papers have reversed them. The results suggest that the models trained on UBFC-rPPG may lack generalizability and perform poorly on PURE. The good news is that our collected dataset can solve this problem. The models trained on the collected dataset can be applied to different test sets.

Table 3: Results on collected dataset

| Method | Entire dataset | | | rPPG subset | | |
|---|---|---|---|---|---|---|
| | MAE↓ | RMSE↓ | $\rho$↑ | MAE↓ | RMSE↓ | $\rho$↑ |
| SEQ-tiny | 3.57 | 6.98 | 0.72 | 5.70 | 9.82 | 0.60 |
| SEQ-T | 1.77 | 4.69 | 0.85 | 3.16 | 6.69 | 0.72 |
| SEQ-FT | **0.95** | **2.12** | **0.96** | **1.55** | 3.29 | 0.89 |
| TS-CAN | 1.39 | 2.47 | 0.95 | 2.44 | 3.60 | 0.86 |
| PhysNet | 3.13 | 6.61 | 0.78 | 4.94 | 9.80 | 0.56 |
| EfficientPhys-C | 1.26 | 2.17 | **0.96** | 1.91 | **2.70** | **0.94** |
| CHROM | 7.73 | 10.8 | 0.43 | 8.50 | 10.0 | 0.12 |
| POS | 3.91 | 6.03 | 0.70 | 5.10 | 7.10 | 0.43 |
| ICA | 11.4 | 13.5 | 0.29 | 12.2 | 14.7 | 0.17 |

Table 4: Cross-dataset evaluation on UBFC-rPPG

| Method | Training set | MAE↓ | RMSE↓ | $\rho$↑ |
|---|---|---|---|---|
| SEQ-tiny | Collected | 0.82 | 1.21 | 0.99 |
| SEQ-tiny | PURE | 6.66 | 9.78 | 0.68 |
| SEQ-T | Collected | 0.93 | 1.47 | 0.99 |
| SEQ-T | PURE | 4.36 | 7.84 | 0.77 |
| SEQ-FT | Collected | 0.81 | 1.21 | 0.99 |
| SEQ-FT | PURE | 2.45 | 4.07 | 0.95 |
| PhysNet | Collected | 0.72 | 1.16 | 0.99 |
| PhysNet | PURE | 1.99 | 4.49 | 0.97 |
| TS-CAN | Collected | 0.69 | 0.95 | 0.99 |
| TS-CAN | PURE | 1.47 | 2.31 | 0.99 |
| EfficientPhys-C | Collected | 0.87 | 1.48 | 0.99 |
| EfficientPhys-C | PURE | 2.04 | 3.06 | 0.99 |

Table 5: Cross-dataset evaluation on PURE

| Method | Training set | MAE↓ | RMSE↓ | $\rho$↑ |
|---|---|---|---|---|
| SEQ-tiny | Collected | 5.80 | 12.0 | 0.86 |
| SEQ-T | Collected | 4.34 | 9.40 | 0.91 |
| SEQ-FT | Collected | 3.55 | 8.86 | 0.92 |
| PhysNet | Collected | 7.83 | 12.1 | 0.89 |
| PhysNet | UBFC | 8.39 | 19.2 | 0.71 |
| TS-CAN | Collected | 3.06 | 6.27 | 0.95 |
| TS-CAN | UBFC | 5.75 | 16.3 | 0.74 |
| EfficientPhys-C | Collected | 1.82 | 3.91 | 0.99 |
| EfficientPhys-C | UBFC | 5.90 | 9.75 | 0.92 |

## 5.4 COMPUTATION COST

Table 6: Computation cost on different platforms

| Method | Batch | Memory usage | | Desktop GPU | | Desktop CPU | | Mobile CPU | |
|---|---|---|---|---|---|---|---|---|---|
| | | Model (KB)↓ | Input (KB)↓ | Time (ms)↓ | Load↑ | Time (ms)↓ | Load↑ | Time (ms)↓ | Load↑ |
| SEQ-tiny | 450 | 13.8 | 346 | 1.1 | 97% | 0.7 | 12% | 6.3 | 40% |
| SEQ-T | 450 | 608 | 346 | 1.7 | 99% | 2.4 | 50% | 34 | 72% |
| SEQ-FT | 450 | 1064 | 346 | 2.2 | 99% | 12 | 17% | 146 | 61% |
| TS-CAN | 128 | 2132 | 1991 | 9.1 | 99% | 79 | 77% | 1260 | 91% |
| PhysNet | 128 | 3080 | 1573 | 9.7 | 99% | 48 | 79% | 1290 | 89% |
| EfficientPhys-C | 128 | 8656 | 7963 | 11.0 | 99% | 271 | 68% | 8530 | 93% |

We also tested the computation cost of the model on different platforms (see Table 6). Unlike the time spent per frame in Liu et al. (2020), we calculated the time spent to process a whole batch. In past work, little attention was paid to the input size. Usually, the input is resized, and a float32 numerical matrix is obtained. If the input size is large, then the memory usage of the input data may be higher than the model itself. The memory usage of the model is a theoretical minimum based on the number of parameters, which will be smaller than the actual value. We tested the average inference time of the model on large data as the time-use result, device load is also concerned. The slow inference speed of the model on some devices may be due to the inability to use computational resources efficiently, resulting in a low load state on the device. Although both low load and high computation cost result in slower speeds, the energy consumed and heat generated are not the same. Smaller computation cost means lower heat generation and more energy efficiency, which is espe-

cially important on battery-powered devices. Our model has a significant advantage in speed, being much faster and consuming much less power than its predecessor. These models can be divided into three classes depending on ARM mobile CPUs' performance. ①: Can run in real-time on desktop platforms but not on mobile devices, such as EfficientPhys. ②: Can run in real-time on mobile devices, but the computation cost is significant and increases heat dissipation and battery loads, such as PhysNet and TS-CAN. ③: Can run smoothly on mobile devices, and the computation cost is tiny, such as the proposed method.

## 5.5 VISUALIZATION & DISCUSSION

To visually verify the effectiveness of the proposed method, SEQ-FT was chosen for visualization, and we randomly selected a video to feed into the model, and the intermediate feature maps were captured and used for visualization whenever possible. Each feature map can be viewed as a multi-channel signal sequence, and its Power Spectral Density (PSD) is calculated for each channel. The 0.5-2.5 Hz portion is intercepted for visualization. A significant heart rate peak is observed in the output from the linear mapping. However, a considerable amount of noise power is also present in the PSD. As the features go deeper, the noise is contained, and some details start to appear, such as subpeaks in the high-frequency region. As shown in Fig 6, the final output PSD is very close to the ground truth. Through feature visualization, our model is well interpreted, and our visualization of temporal modeling demonstrates the role of convolution networks in another dimension than Liu et al. (2020) and Niu et al. (2020a)'s focus on spatial modeling.

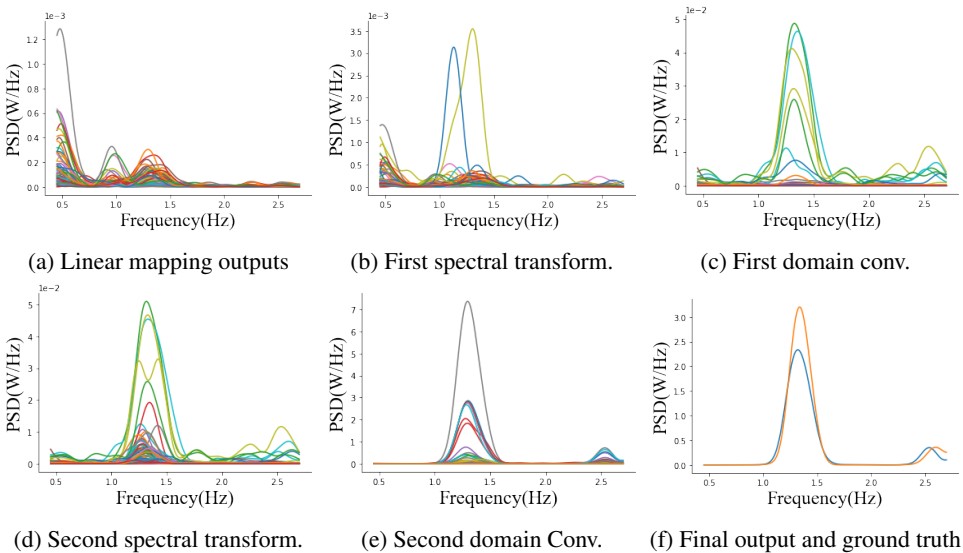

Figure 6: Power Spectral Density(PSD) of the feature maps.

## 6 CONCLUSIONS

In terms of accuracy, our model performs similarly to or better than existing models; in terms of speed, our model reaches several times more than current models and can run effectively on mobile CPUs. On Raspberry Pi 4B, our highest accuracy model inference takes 146ms, and the fastest model takes 6.3ms, compared with TS-CAN's 1260ms and EfficientPhys' 8530ms. However, SEQ-FT does not perform as expected on CPUs, and it is worth investigating how to make the spectral transform run more efficiently on CPUs. The quality of our dataset is better than UBFC-rPPG and PURE. The accuracy of existing models can be improved significantly by training on the collected dataset, e.g., the MAE of TS-CAN decreases from 1.47 (trained on PURE) to 0.69; the MAE of PhysNet decreases from 1.99 (trained on PURE) to 0.72, and we expect the collected dataset to be a baseline for training sets. However, the proposed method is sensitive to the random shift present in the dataset, and we will try to deal with this issue in the future.

## 7  REPRODUCIBILITY STATEMENT

Although the steps required to reproduce the results have been described in this paper, factors that significantly affect the results need to be highlighted in this section. This model has not been finely tuned to the parameters, and its accuracy is not very sensitive to the parameters but to the training set. When working with datasets, there were some key operations: First, we implemented face detection and tracking. However, face detection frames are usually difficult to keep stable. We added a displacement threshold of size 25 pixels when implementing face tracking. Any movement at a distance less than the threshold will be filtered, and greater than the threshold will move the detection frame suddenly, thus avoiding introducing periodic noises. Secondly, when making the training set, we made additional high heart rate samples with heart rates in the 120-180, representing about 25% of the total number of samples. The samples were obtained by skipping half of the frames at intervals, so the heart rate would be twice as high, which is exceptionally important for training the model to correctly identify the high heart rate samples; if enough high heart rate samples are not correctly generated in the training set, then great errors may occur by chance when estimating the high heart rate test set samples. Finally, we ensured that the BVP signal in the training set was strictly aligned with the frames by the recorded UNIX timestamps. Our model does not require facial segmentation, but tracking using facial detection is necessary due to the uncertainty of the head position in the collected dataset. The above steps were applied to all models we implemented.

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

# A APPENDIX

## A.1 PARAMETER DISCUSSION AND ABLATION EXPERIMENTS

Our approach is sensitive to the length of the input frame sequence due to its focus on modeling in the time dimension. We trained SEQ-FT on the collected dataset and tested it on the collected dataset and UBFC-rPPG(Table 7). From the results, a longer sequence can improve the accuracy. However, we do not want too long input sequences to cause too much delay, so 15 seconds(450 frames) is a suitable value.

Table 7: Effect of different input sequence lengths on accuracy

| Length | Mean Absolute Error↓ | |
| --- | --- | --- |
| | Collected | UBFC-rPPG |
| 128 | 3.22 | 12.6 |
| 256 | 2.21 | 3.24 |
| 384 | 1.09 | 1.57 |
| 450 | 0.95 | 0.81 |
| 512 | **0.81** | **0.77** |

Our model ignores spatial information, so a higher resolution does not help to improve the accuracy, but greatly increases the parameters. We trained SEQ-FT on the collected dataset, and tested on the collected dataset and UBFC-rPPG(Table 8). The results show that using a resolution of 8×8 is sufficient.

Table 8: The effect of different resolutions on accuracy

| Resolution | Mean Absolute Error↓ | |
| --- | --- | --- |
| | Collected | UBFC-rPPG |
| 4×4 | 3.34 | 7.49 |
| 8×8 | **0.95** | **0.81** |
| 16×16 | 0.96 | 0.98 |
| 32×32 | 1.05 | 0.92 |

Our model uses time-domain convolution and spectral transformation, and ablation experiments can show that the combination of these two is optimal. All models were trained on the collected dataset, and tested on the collected dataset and UBFC-rPPG.(Table 9)

## A.2 PRIVACY

If the rPPG method is applied to a commercial application, the server expects to capture the user's facial image to optimize the user experience and improve the algorithm. However, the user's facial

Table 9: Effect of different model structure and number of layers on accuracy

| Layers | MAE on collected dataset↓ | | | MAE on UBFC-rPPG↓ | | |
|---|---|---|---|---|---|---|
| | F[1] | T[2] | F[1]&T[2] | F[1] | T[2] | F[1]&T[2] |
| 1 | 3.73 | 3.85 | - | 2.61 | 7.81 | - |
| 2 | 2.84 | 1.86 | 1.60 | 0.97 | 3.74 | 2.14 |
| 3 | 2.18 | 1.80 | - | 1.08 | 2.31 | - |
| 4 | 1.77 | 1.44 | **0.95** | 0.93 | 1.90 | **0.81** |
| 5 | 1.40 | 1.43 | - | 15.3 | 1.45 | - |
| 6 | 1.65 | 1.21 | 1.08 | 14.8 | 1.32 | 1.49 |

[1] Spectral transformation in the frequency domain
[2] 1D convolution in the time domain

image involves privacy issues, and the higher resolution, the more private information it contains (see Fig 7), so using low-resolution images helps to protect privacy. Besides, low-resolution images can reduce network usage during uploading and improve the user experience. Our model uses 8x8 inputs, which implies strong privacy.

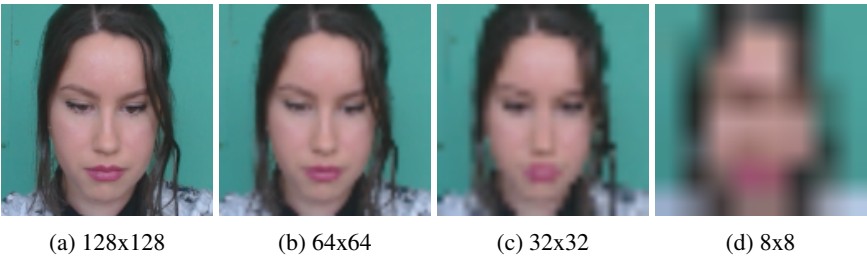

| (a) 128x128 | (b) 64x64 | (c) 32x32 | (d) 8x8 |

Figure 7: Facial images in different resolutions.

### A.3 Low Latency

Since our model uses a long sequence of 450 frames of input, it usually requires 15 seconds to obtain the BVP signal. Compared with models using 128 frames of input, our model seems to have to wait longer. However, in engineering, this can be optimized. For example, for a 60 frames input, we simply repeat it eight times to get 480 frames of input and intercept 450 of them. In the output signal, the BVP waveform can be obtained by intercepting 60 frames from the middle at the corresponding position, and this operation greatly reduces the delay (see Fig 8).

### A.4 Signal Quality

TS-CAN must use a bandpass filter to improve signal quality. For PhysNet, additional filters are not required. the use of additional filters has almost no effect on SEQ-FT (see Fig 9).

### A.5 More Efficient Pre-processing

Simple pre-processing can be eliminated by reasonable engineering optimization. In the original article, the pre-processing of TS-CAN was done by NumPy, and we tried to do the same using Tensorflow and move this operation inside the model. This means that we only need to input the original video after deploying the model and do not need to pay attention to the specific implementation of pre-processing, which helps it to be implemented in non-python environments (e.g. mobile or web), this operation does not affect the accuracy and makes full use of Tensorflow's powerful matrix operations.

```
# Approach of the original literature
motion, appearance = pre_process(vid) # Pre-processing with numpy
```

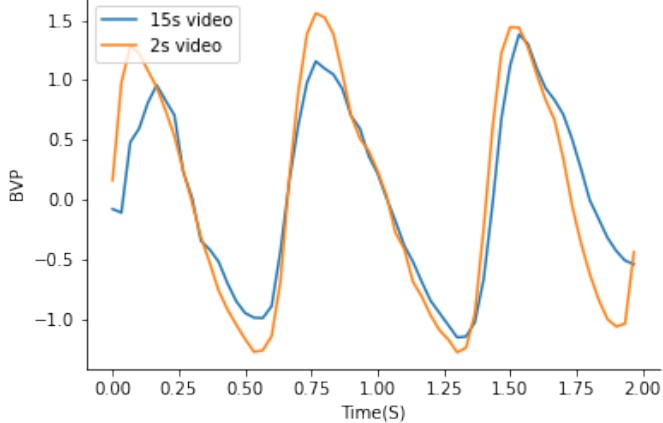

Figure 8: Extract the BVP signal from 2 seconds or 15 seconds of video in the same duration

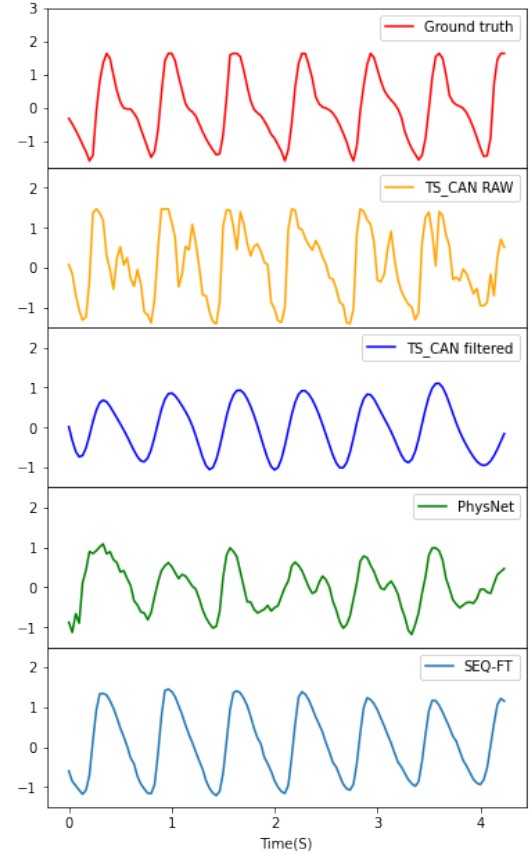

Figure 9: BVP Signal from different model

```
BVP = TS_CAN([motion, appearance])
```

```
.......
   def call(self, x):
       x_ = (x[1:] - x[:-1])/(x[1:]+x[:-1])
```

```
        x_ = (x_ - tf.reduce_mean(x_, axis=(0, )))/tf.math.reduce_std(x_,
            axis=(0, ))
        motion, appearance = tf.concat([x_, tf.zeros([1, *size, 3])],
            axis=0), tf.expand_dims(tf.reduce_mean(x, axis=(0, )), axis=0)
        d1, r1 = self.diff_input(motion), self.rawf_input(appearance)
        .........
# A better approach, closer to end-to-end
BVP = TS_CAN(vid)
```

