# OpenReview forum: "SEQuence-rPPG: A Fast BVP Signal Extraction Method From Frame Sequences"
_ICLR.cc/2023/Conference — Submitted to ICLR 2023_

### Official Review · Reviewer_V5oM · 2022-10-24

**Confidence:** 3
**Correctness:** 3
**Technical Novelty And Significance:** 2
**Empirical Novelty And Significance:** 2
**Recommendation:** 3

**Clarity, Quality, Novelty And Reproducibility:**

As mentioned above it is not easy to follow the results discussion. The paper also needs proofreading. There are several typos and sentences which need to be rewritten. A non-exhaustive list from the introduction only is the following (but all sections need proof-reading):

(ICA), In Poh et al. (2010b), -> (ICA). In Poh
new color space and separates -> …which separates
separate present -> maybe separately present (?)
facial image segmentation can limit its deployment on mobile devices because it is currently difficult to achieve on mobile devices effectively. -> this sentence needs to be rewritten
movements. the large size -> movements. The large

The authors provide several details about their method but still the results are not fully reproducible since the training and test sets are generated randomly.


**Strength And Weaknesses:**

Strengths

A new dataset is presented which will be a nice contribution (if it becomes available).

Weaknesses

A major weakness in the paper is that it’s not easy to follow the discussion. There are no references to tables or figures, so the reader is left searching for the corresponding tables/figures.

The introduction reads more like a review of existing methodologies, it’s hard for the reader to understand what the main contributions are since they are described in different paragraphs. It would be good to have a paragraph which lists the main contributions. For example the end of the last paragraph could be extended to summarise all contributions.

An ablation study is missing in the paper. This would reveal the impact of each component.
For example, the authors use an image of 8 by 8 in order to alleviate privacy concerns. However, this is a very low resolution and it would be good to test the performance of the proposed approach as a function of the input resolution. Another example is studying the impact of the number of convolutional and spectral layers and the impact of the input sequence length.

One of the main contributions of this paper is the introduction of a new dataset, however only a brief description of the dataset is given and it is not mentioned if the dataset will become publicly available. The latter Is quite important as if the dataset is not released it diminishes the paper’s contribution. It would also be useful to add more details in the dataset description, e.g., age/gender distribution, did all subjects do all tasks, total number of videos and duration of the entire dataset.


**Summary Of The Paper:**

This paper introduces a new dataset for blood volume pulse (BVP) estimation which addresses some shortcomings of existing datasets. A method to estimate BVP from RGB frames is also introduced. Results on the collected dataset and other publicly available datasets are presented and the proposed approach achieves competitive performance without requiring complex pre-processing.



**Summary Of The Review:**

There are several weaknesses as explained above which need to be addressed before the paper can be considered for acceptance.

---

> ### Author Response · Authors · 2022-11-19
> **Thank you for your useful advice!**
>
> We thank the reviewers for their suggestions, which were very helpful. We have been able to improve our work in the following ways.
>
> >There are no references to tables or figures, so the reader is left searching for the corresponding tables/figures.
>
> We are revising the paper to make it as readable as possible.
>
> >It would be good to have a paragraph which lists the main contributions.
>
> Thank you for your suggestion, we have adjusted the structure of the paper.
>
> >An ablation study is missing in the paper. This would reveal the impact of each component.
>
> We added the ablation study, see Appendix A.1.
>
> >however only a brief description of the dataset is given and it is not mentioned if the dataset will become publicly available.
>
> We added some information about the dataset and included an example in the supplementary material. Although the diversity of this dataset is limited, its duration is over 30 hours (3.24M frames)
>
> >As mentioned above it is not easy to follow the results discussion. The paper also needs proofreading
>
> The preliminary proofread version has been updated, please check.
>
> >still the results are not fully reproducible since the training and test sets are generated randomly.
>
> The training and test sets are randomly generated, but we have saved them, please check the supplementary material.

---

> > ### Comment · Reviewer_V5oM · 2022-11-28
> > **Reply**
> >
> >
> > I would like to thank the authors for their reply. However, most of my concerns about the manuscript remain still remain. It's also not easy to find out what changes were made to the paper by the authors. Finally, I still don't fully understand how it's possible to estimate BVP from an 8x8 image when it's so heavily distorted (as shown in the appendix).

---

### Official Review · Reviewer_j11H · 2022-10-25

**Confidence:** 5
**Correctness:** 3
**Technical Novelty And Significance:** 1
**Empirical Novelty And Significance:** 1
**Recommendation:** 3

**Clarity, Quality, Novelty And Reproducibility:**

The work requires polishing and editing, and perhaps a re-write in a few areas. The contributions are not clear. The methodology is trivial, and many details are missing.

**Details Of Ethics Concerns:**

Given that the work includes new human-related data, it is not clearly mentioned if ethics approval has been secured or not.

**Strength And Weaknesses:**

The main strength of the paper is the dataset component of the work. However, the paper, unfortunately, has a number of weaknesses that needs to be addressed. These weaknesses are as follows:

1- The writing, presentation, and editing are not very polished.

2- The paper is very hard to follow. I find it hard to put my finger on what exactly the contributions of the paper are.

3- The proposed method is very simple and trivial, and does not provide any new "representation learning" insights.

4- The paper cites several well-known datasets in the area, including MAHNOB-HCI, VIPL-HR, and others, some of which have more participants and are more diverse. It is therefore not clear why a new dataset was needed.

5- The dataset is not very diverse in terms of participants, and many details are missing, e.g. M/F ratio, age range/SD, etc

6- There is no confirmation that ethics approval has been secure, which is really important.

7- The dataset is not made public (and doesn't seem to be planned to go public). So I am not sure if it can be considered a contribution to the area.

**Summary Of The Paper:**

The paper proposes a simple method, followed by a dataset for remote PPG/BVP monitoring. The proposed method is a simple neural model with reasonable results, and the dataset has 58 participants.

**Summary Of The Review:**

Please see my review above.

---

> ### Author Response · Authors · 2022-11-19
> **Thank you for taking the time to read the paper.**
>
> We thank the reviewers for their criticism of the paper, which helped us to improve this paper. Specifically, there are the following points.
>
> >1- The writing, presentation, and editing are not very polished.
>
> Sorry, we are in the process of revising.
>
> >2- The paper is very hard to follow. I find it hard to put my finger on what exactly the contributions of the paper are.
>
> We restructured the paper to clarify its contribution.
>
> >3- The proposed method is very simple and trivial, and does not provide any new "representation learning" insights.
>
> Simplicity is a prerequisite for reliability and developability, and our approach is based on a priori knowledge in the domain. There are simple and effective methods based on deep learning that have underperformed in the past because there is not yet a suitable training set.
>
> >It is therefore not clear why a new dataset was needed.
>
> Many models are sensitive to datasets, limiting the community due to the poor synchronization and limited size of most datasets. See Table 4 and Table 5, our training set can improve the performance.
>
> >5- The dataset is not very diverse in terms of participants, and many details are missing, e.g. M/F ratio, age range/SD, etc
>
> We have added some information. Since the dataset was used for education and therefore the subjects were almost exclusively master's degree students, we acknowledge that sample diversity is limited.
>
> >6- There is no confirmation that ethics approval has been secure, which is really important.
>
> We are well prepared for ethical review to ensure that datasets can be released.
>
> >7- The dataset is not made public (and doesn't seem to be planned to go public). So I am not sure if it can be considered a contribution to the area.
>
> This dataset will be made public.

---

> > ### Comment · Reviewer_j11H · 2022-11-28
> > **Reply**
> >
> > I appreciate the author's efforts in improving the work. However, unfortunately my concerns about the paper (notably lack of key contributions w.r.t. representation learning, dataset, etc) persist. I therefore think the paper doesn't meet the standards of ICLR. So I will opt to keep my original score. I wish the authors the best in improving the paper and getting it published in another venue.

---

### Official Review · Reviewer_DUL3 · 2022-10-25

**Confidence:** 4
**Correctness:** 3
**Technical Novelty And Significance:** 3
**Empirical Novelty And Significance:** 2
**Recommendation:** 6

**Clarity, Quality, Novelty And Reproducibility:**

I have some concerns and the reproducibility and as a related topic, to the comprehensiveness of the results, as described  in the previous section.

**Details Of Ethics Concerns:**

I don't have ethical concens per se.  However, I did question above whether it would be possible for the author to release the data used and this would be a question related to their IRB.  I would like to know the plans, as the authors make the dataset a central part of the story in this paper.

**Strength And Weaknesses:**

The authors correctly highlight deficiencies with the existing public datasets which are invariably small and limited in diversity. They present a new dataset that they collected which is arguably larger than the existing data. However, their dataset is also limited in diversity with regard to skin types and age, based on the description. I think their claims that their dataset is particularly large in comparison to existing datasets is not really true - it is comparable in order of magnitude. I think that those claims should be tempered a bit. It was not very clear to me whether the dataset will be made available as a benchmark set?  Please clarify - making the contribution of the dataset a central claim in the paper without releasing it is frustrating to readers as it doesn’t help to advance the communities’ efforts, beyond the insights gained from a single paper.

The authors state that for some existing methods the "input batch size is 128 frames, which means it has a latency of over 1s. To get a real-time BVP signal, the latency should be as small as possible." However, it is not that clear to me in what application a delay of 1 or even a couple of seconds would be a problem.  In heart rate estimation a period of at least 5 seconds would be need at least any way.  Can the authors clarify why a latency of 1 second is significant here?  I can see the memory or compute constraints with a longer temporal window being more significant which may better support the authors arguments here.

The approach to solving the problem does appear to be novel.  It is interesting that the authors collapse the input to RGB vectors thereby losing spatial information, which is often useful for segmentation.  I would question how sensitive their method is the any face detection or segmentation step they perform. I would appreciate a comment on this, again with little visibility on the dataset they collected it is hard for a reader to ascertain this.  To add to this point, the paper is short on experimental details and while the supplementary material contains model implementations, it does not actually include end-to-end training code which is disappointing as their are often details in steps prior to, or following, a model that impact performance (e.g., segmentation steps, filtering steps etc.). I would appreciate the authors comments on any additional details that could be provided to help ensure reproducibility.

Section 5:  It is still a bit unclear to me after reviewing the paper why the proposed method cannot be trained on other datasets.  Yes, some datasets do not have very highly synchronized PPG and video, but some do.  In my experience, for example, training models with PURE or SCAMPS is possible.  Can the authors be more precise about why training with those existing datasets isn’t possible? I would appreciate a clearer indication or examples to illustrate this.
And in Table 4 - why wasn’t EfficientPhys, PulseGAN or DualGAN trained on the collected dataset?  Is that because those models were not implemented and the results taken from prior work?

The above point applied to training, for testing it would seem that the authors could use any dataset.  Why did the authors only choose to test on UBFC and not any of the other benchmark datasets?  This is a significant weakness of the work as almost all other papers manage to test on at least 2-3 public datasets.  Could the authors produce results on these other datasets?

In Section 5 the authors write: Unlike the time spent per frame in Liu et al. (2020), we calculated the time spent to process a whole batch, which is more reflective of the latency caused by the computation. However, while this is true about latency, it doesn’t reflect the computational cost which I think is more the point (see the comment above about the difference between latency and computational cost). I don’t think latency is really a bottleneck in this application, but computation is.


**Summary Of The Paper:**

This paper presents an approach for extracting the blood volume pulse signal via photoplethysmography from a video.  This use case of computer vision has several positive applications, that could help make scalable physiological sensing possible.  The authors do a thorough job summarizing the existing literature, including many of the most recent papers.  While the summary could be made a little more readable, I think it is a well synthesized introduction to the paper on whole.  The proposed method uses existing but computationally efficient neural layers to extract a PPG signal from the video.  The results overall appear reasonable and the method performs well compared to the baselines in the comparisons that are given.


**Summary Of The Review:**

Overall, I believe this application is very exciting.  I think that the proposed method has some novelty and focusing on fast/computationally light algorithms is good.  The paper has the makings of a good contribution that I’d like to support.  However, I feel like the paper as it is currently falls short in empirical support for their method and theoretical support. If the authors can address all these points above - I would be willing to increase my rating.

---

> ### Author Response · Authors · 2022-11-19
> **Thank you for your thorough review of our paper, we addressed some points.**
>
> We thank the reviewers for their comments on the paper. The paper has some shortcomings and some parts are still unclear, and we have made additions as follows.
>
> >I think their claims that their dataset is particularly large in comparison to existing datasets is not really true
>
> Yes, our dataset is inadequate in terms of diversity, because this is a dataset dedicated to the education.  What we claim, that it is very large, refers to the duration, it contains more than 30 hours (3.24M frames) of video, larger than any public dataset we know.  More frames help model training.
>
> >It was not very clear to me whether the dataset will be made available as a benchmark set? Please clarify
>
> Yes, the dataset will be made public.
>
> > Can the authors clarify why a latency of 1 second is significant here?
>
> We have changed the wording. In most cases the latency depends on the computation cost, and we expect the computation cost to be as low as possible.
>
> > I would question how sensitive their method is the any face detection or segmentation step they perform.
>
> On UBFC-rPPG and PURE dataset, face detection is not required. We had to use the face detection because our subjects needed about 45 minutes to complete the task, so we did not constrain the head movements, and there may have been relatively large head movements.
>
> >again with little visibility on the dataset they collected it is hard for a reader to ascertain this
>
> Please check the new supplementary material, we have posted a preview sample.
>
> >it does not actually include end-to-end training code
>
> Our work is still in progress, the code is still being organized, once it is finished, we will release it on github.
>
> >Can the authors be more precise about why training with those existing datasets isn’t possible?
>
> Our model requires a highly synchronized training set, we are sorry that we were not aware that the PURE dataset could be used for training, the relevant experimental results have been published in the revised version. The synthetic dataset is a great idea and we are starting work on it, we have concerns about the synthetic dataset because, according to the results given by the authors[1], it does not generalize very well. We also focus on synthetic datasets and have made some attempts, and so far it is difficult to see how it can surpass the collected dataset.
>
> >why wasn’t EfficientPhys, PulseGAN or DualGAN trained on the collected dataset?
>
> We focus on small end-to-end models, PulseGAN and DualGAN require complex pre-processing and therefore are not used for comparison. EfficientPhys contains both convolution-based and Transformer-based models, and according to the paper, we reproduced EfficientPhys-C, using 72*72 resolution, according to the paper, this is the best performing structure. In our experiments it performs well, but the computation cost is much larger than TS-CAN and our model and cannot run in real time on ARM CPUs.
>
> >Why did the authors only choose to test on UBFC and not any of the other benchmark datasets?
>
> We supplemented the experiments. Results on the PURE dataset have been released.
>
> >I don’t think latency is really a bottleneck in this application, but computation is.
>
> I agree. we don't use the average frame time cost because our method is very fast and it uses a very long frame sequence input. Its average time cost per frame is less than 1ms on mobile CPUs, which may be not a fair comparison, we use latency to avoid potentially misleading readers.
>
> [1] Liu et.al., Deep Physiological Sensing Toolbox.

---

> > ### Comment · Reviewer_DUL3 · 2022-11-23
> > **Thank you for your response.**
> >
> > I would like to thank the authors for their responses.  I appreciate the additional evidence and experiments.
> > It seems there are some weaknesses in the model's performance compared to the baselines, accepting that some of the baselines are more computationally expensive, but models such as TS-CAN are relatively fast. Overall, I feel the work doesn't provide a highly compelling case for their architecture; however I do believe that their dataset would be valuable to the community. The paper could be improved in clarify and presentation overall.  I think that some of these things are addressable in a revision.  I am willing to increase my rating to 6 in recognition for providing these extra results and the release of the dataset. However, I am not completely convinced that this paper is clearly over the bar for ICLR and it seems that the other reviewers might agree with this assessment. I think the authors might be best suited revising the paper it to make it stronger for a future venue and preparing the dataset release ahead of the submission that it can be fully considered in the evaluation of the submission.

---

### Author Response · Authors · 2022-11-19
**The paper has been revised**

Thank you to all reviewers for their comments! All comments have been read carefully and are summarized below based on the issues that were of most interest.

**1-The dataset will be made public.**
We hope to contribute to the rPPG community and this will be a good training set.  Many datasets exist in the community for evaluating models, but not all of them are suitable for training models, and our datasets are designed for training.


**2-Some new experimental results were added to the paper.**
We added a series of experiments, including cross-dataset evaluation on the pure dataset, replication of EfficientPhys-C, ablation study.
The experiments show that the proposed method is competitive with larger models in terms of accuracy. All models can be trained on the collected dataset and improve the accuracy.


**3-The text has been revised to highlight the contribution of the paper.**
The introduction section has been revised to help the reader understand the paper's contributions more clearly.

---

### Decision · Program_Chairs · 2023-01-20

**Decision:**

Reject

**Justification For Why Not Higher Score:**

N/A

**Justification For Why Not Lower Score:**

N/A

**Metareview: Summary, Strengths And Weaknesses:**

This paper proposed a new method for extracting the blood volume pulse signal via photoplethysmography from a video, which is simple, fast and accurate. While the proposed technique may be practically useful, reviewers raised major weakness concerns about the incremental contributions, limited novelty, weak experimental results and lack of clarity in the presentation. While the authors tried to give responses to address the review questions, the overall quality of this work is still below the acceptance bar.